# Repeated Infusions of Bone-Marrow-Derived Mesenchymal Stem Cells over 8 Weeks for Steroid-Refractory Chronic Graft-versus-Host Disease: A Prospective, Phase I/II Clinical Study

**DOI:** 10.3390/ijms25126731

**Published:** 2024-06-19

**Authors:** Nayoun Kim, Gi-June Min, Keon-Il Im, Young-Sun Nam, Yunejin Song, Jun-Seok Lee, Eun-Jee Oh, Nack-Gyun Chung, Young-Woo Jeon, Jong Wook Lee, Seok-Goo Cho

**Affiliations:** 1Institute for Translational Research and Molecular Imaging, College of Medicine, The Catholic University of Korea, Seoul 06591, Republic of Korea; nkim@lucasbio.com (N.K.); beichest@catholic.ac.kr (G.-J.M.); keonil1200@gmail.com (K.-I.I.); youngsun.nam@thermofisher.com (Y.-S.N.); yjsong@seegene.com (Y.S.); poi03005@catholic.ac.kr (J.-S.L.); 2Department of Hematology, Seoul St. Mary’s Hematology Hospital, College of Medicine, The Catholic University of Korea, Seoul 06591, Republic of Korea; cngped@catholic.ac.kr (N.-G.C.); jwlee@catholic.ac.kr (J.W.L.); 3Department of Laboratory Medicine, Seoul St. Mary’s Hospital, College of Medicine, The Catholic University of Korea, Seoul 06591, Republic of Korea; ejoh@catholic.ac.kr; 4Department of Hematology, Yeouido St. Mary’s Hospital, College of Medicine, The Catholic University of Korea, Seoul 06591, Republic of Korea; native47@catholic.ac.kr

**Keywords:** mesenchymal stem cells, mesenchymal stromal cells, chronic graft-versus-host disease, clinical trial, cell therapy

## Abstract

Chronic graft-versus-host disease (cGVHD) is a long-term complication of allogeneic hematopoietic stem cell transplantation associated with poor quality of life and increased morbidity and mortality. Currently, there are several approved treatments for patients who do not respond to steroids, such as ruxolitinib. Nevertheless, a significant proportion of patients fail second-line treatment, indicating the need for novel approaches. Mesenchymal stem cells (MSCs) have been considered a potential treatment approach for steroid-refractory cGVHD. To evaluate the safety and efficacy of repeated infusions of MSCs, we administered intravenous MSCs every two weeks to ten patients with severe steroid-refractory cGVHD in a prospective phase I clinical trial. Each patient received a total of four doses, with each dose containing 1 × 10^6^ cells/kg body weight from the same donor and same passage. Patients were assessed for their response to treatment using the 2014 National Institutes of Health (NIH) response criteria during each visit. Ten patients with diverse organ involvement were enrolled, collectively undergoing 40 infusions as planned. Remarkably, the MSC infusions were well tolerated without severe adverse events. Eight weeks after the initial MSC infusion, all ten patients showed partial responses characterized by the amelioration of clinical symptoms and enhancement of their quality of life. The overall response rate was 60%, with a complete response rate of 20% and a partial response (PR) rate of 40% at the last follow-up. Overall survival was 80%, with a median follow-up of 381 days. Two patients died due to relapse of their primary disease. Immunological analyses revealed a reduction in inflammatory markers, including Suppression of Tumorigenicity 2 (ST2), C-X-C motif chemokine ligand (CXCL)10, and Secreted phosphoprotein 1(SPP1), following the MSC treatment. Repeated MSC infusions proved to be both feasible and safe, and they may be an effective salvage therapy in patients with steroid-refractory cGVHD. Further large-scale clinical studies with long-term follow-up are needed in the future to determine the role of MSCs in cGVHD.

## 1. Introduction

Graft-versus-host disease (GVHD) is a serious complication that is caused by allogeneic hematopoietic cell transplantation (HCT) [1]. Advances in conditioning regimens and post-transplant care have significantly enhanced overall survival following HCT. However, as more patients achieve long-term survival, chronic GVHD (cGHVD) emerges as a significant threat that diminishes their quality of life. Corticosteroids are the primary treatment choice for cGVHD, but only half of patients respond to treatment. For the remaining half, multiple drugs and approaches have been approved as second-line treatment [2,3], including mTOR inhibitors [4], nilotinib [5], imatinib [6], extracorporeal photopheresis (ECP) [7], ruxolitinib [8], ibrutinib [9,10], and belumosudil [11].

Bone-marrow-derived mesenchymal stromal cells (MSCs) have been widely applied in clinical trials due to their immunomodulatory properties [12,13,14,15]. Clinical applications of MSCs include, but are not limited to, acute respiratory distress syndrome [16], Crohn’s disease [17], type 1 diabetes [18], osteoarthritis [19], multiple sclerosis [20], and liver diseases [21]. MSCs have predominantly been used in the treatment of acute GVHD (aGVHD) patients ever since the first clinical study of their use [22]. In contrast, published data on the use of MSCs as a therapeutic option for patients with cGVHD remain limited [13]. cGVHD, in contrast to aGVHD, shows a distinct pathophysiological nature characterized by autoimmune-like mechanisms, chronic inflammation, and the subsequent development of fibrosis and scarring. Over the years, there has been widespread acceptance of the notion that the immunoregulatory properties of MSCs are not an inherent trait but rather are activated or enhanced within an appropriate inflammatory microenvironment [23,24,25]. These observations have led to extensive investigations into enhancing the therapeutic efficacy of MSCs through genetic modification and pre-conditioning techniques [25,26]. Our research group, as well as several others, have shown evidence that a single dose of MSCs is not sufficient for therapeutic efficacy and that multiple administrations of MSCs can amplify their therapeutic effects in preclinical and clinical studies [27,28]. However, the application of multiple infusions of MSCs in the clinical setting may have some limitations. Additional doses of MSCs have been mainly administered on an ‘ad hoc’ basis and given primarily when MSCs are accessible or when a patient fails to exhibit a clinical response to the previous single injection. Consequently, the number of doses administered varies significantly within a single clinical study [29]. 

In this clinical trial, our strategy entails the planned administration of multiple doses of MSCs from the point of enrollment. We hypothesized that repeated doses may enable MSCs to exert sustained immunomodulatory properties and prolonged persistence in vivo, ultimately resulting in the amelioration of clinical symptoms in cGVHD patients. Here, we demonstrate the safety and feasibility of such repeated MSC administrations, offering promising potential benefits for cGVHD patients. 

## 2. Results

### 2.1. Repeated Infusions of MSCs Are Safe and Tolerable in Refractory cGVHD Patients 

Ten patients who were diagnosed with chronic graft-versus-host disease (cGVHD) following allogeneic hematopoietic cell transplantation (HCT) were enrolled in this study, as described in Table 1. All patients received HCT for hematologic malignancies and developed either moderate or severe cGVHD involving multiple organs.

The enrolled patients received a total of 40 doses of MSCs, and each dose was administered intravenously over 30 min. The MSCs were prepared fresh one day prior to infusion and were released following the completion of all the release tests. Two MSC batches from each donor were sufficient to cover all 40 doses. All the patients received a cell dose containing 1 × 10^6^ cells/kg at passage 5 at a two-week interval for a total of four doses (Table 2). The HLA matching degree of the infused MSCs with the recipients varied from no matching at all up to a maximum of 6 out of 8 matches. To determine the possibility of the development of anti-HLA antibodies following repeated MSC infusions, we examined the presence of anti-HLA class I, class II, and MIC antibodies in the patients’ serum. We compared the presence of anti-HLA antibodies prior to MSC infusion and 2 weeks following all four MSC administrations. Two patients developed HLA class I antibodies, and one patient developed HLA class II antibodies. However, we did not further observe donor- or MSC-specific antibodies through single-antigen assays. 

Overall, there were no serious adverse events noted during the four repeated infusions of MSCs in each patient. However, three patients (UPN-2, UPN-3, and UPN-9) experienced incidents of fever, all of which resolved without requiring further intervention. UPN-4 showed a grade 2 upper respiratory infection one month after the final MSC infusion, and UPN-8 showed grade 2 diarrhea at the second MSC infusion. Grade 3 or 4 adverse events were not observed in any of the patients, suggesting the safety and tolerability of repeated MSC infusions. All the events are summarized in Appendix A. 

### 2.2. MSC Induces Long-Term Clinical Response in Majority of cGVHD Patients

Following the four doses of MSC infusions, the patients were evaluated for clinical response based on the NIH criteria (Table 3). Overall survival was 80%, with a median follow-up of 381 days (range: 122–397), in which two patients (UPN-5 and UPN-6) had died due to relapse of the primary disease. Furthermore, 80% showed a partial response, and the overall response rate was 80%, indicating an improvement in clinical symptoms. The median time to initial response was 57 days (range: 54–63). Patient 8 showed a mixed response at every response assessment, in which the patient showed improvement in the oral mucosa and joints but deterioration in the skin. Of the eight responders who responded at 8 weeks, five (62.5%) continued to show a sustained response up to 54 weeks. At the last follow-up, which was approximately one year after the first infusion, the overall response rate was 60%, with a complete response rate of 20% and a PR rate of 40%. While the majority of responders showed evidence of a response at the first response assessment at 8 weeks, UPN-3 showed an initial response at 18 weeks and eventually showed a complete resolution of disease at the 42-week assessment.

In Figure 1, we show a representative clinical outcome of UPN-1 prior to (Figure 1A) and following the MSC infusions (Figure 1B). UPN-1 was a 51-year-old male patient who was primarily diagnosed with diffuse large B cell lymphoma and received allogeneic hematopoietic stem cell transplantation from a matched sibling donor. Approximately 4.9 months after transplantation, the patient developed cGVHD and had undergone three lines of therapy by the time of the MSC infusion. The patient exhibited severe cGVHD, with oral, skin, and musculoskeletal involvement. Clinically, he presented skin manifestations reminiscent of lichen sclerosus, accompanied by profound sclerotic eruptions on the soles of his feet, leading to considerable pain and discomfort, necessitating the use of a wheelchair for mobility. Additionally, depigmentation and sclerosis were observed on the face, along with hair loss. Following the repeated MSC infusions, a notable improvement in cutaneous symptoms was observed. Importantly, the patient regained the ability to ambulate on his own at the time of discharge.

Moreover, individual organ responses were observed (Table 4). An organ analysis revealed that the mouth (9/10), eyes (8/10), and joints (6/10) were the most involved organs at the time of infusion. During the initial response assessment at 8 weeks following the first infusion, the mouth and joints were the most responsive organs, showing 78% and 83% response rates, respectively. Furthermore, the response rates were sustained and improved in the mouth and eyes, showing 89% and 75% response rates, respectively. However, three patients showed progressive disease in the skin and the joints, in which the patients developed new symptoms, or pre-existing symptoms worsened. Overall, a well-sustained and long-term clinical response was observed.

Conventional immunosuppressive agents were allowed throughout the study, and the patients were able to taper off immunosuppressants based on the investigator’s decision. It was observed that 4 out of 10 patients had discontinued all immunosuppressants at the last follow-up, and 2 out of 10 patients had discontinued prednisolone but continued to use other agents (Table 5). 

### 2.3. Laboratory Parameters 

The patients were serially monitored for laboratory parameters through complete blood count (CBC) and blood chemistry tests before and following the MSC infusions (Appendix A). There was no significant difference in any of the parameters between the responders and non-responders prior to MSC infusion. At the 8-week follow-up, albumin was significantly reduced in the responders compared to in the non-responders (*p* = 0.044). However, we did notice a reduction in CRP levels in the responders compared to in the non-responders throughout the monitoring period. Overall, the cGVHD patients prior to infusion had elevated levels of aspartate aminotransferase (AST; mean concentration of 3.68 units/L in healthy individuals and 39.4 units/L in cGVHD patients) and alanine transaminase (ALT; mean concentration of 16.17 units/L in healthy individuals and 50.2 units/L in cGVHD patients) and decreased levels of total protein (mean concentration of 7.12 g/dL in healthy individuals and 6.3 g/dL in cGVHD patients) compared to healthy adults (Appendix A). Following the MSC infusions, AST and ALT levels decreased, while the total protein increased, indicating clinical improvement. 

### 2.4. Biomarker Analyses

Next, we analyzed potential biomarkers that may be involved in cGVHD and in determining clinical responses. Out of the 32 markers that we analyzed, we were able to identify 9 that were significantly upregulated in the cGVHD patients before the MSC infusions compared to in healthy individuals (Figure 2). The identified markers were involved in pro-inflammation, and, therefore, we evaluated whether there was any reduction in their levels following the MSC infusions (Figure 3). While there were no statistically significant changes over time, markers such as ST2 (median concentration of 1396.27 pg/mL at 0 weeks and 363.92 pg/mL at 54 weeks) and IL-1Rα (median concentration of 247.74 pg/mL at 0 weeks and 0 pg/mL at 54 weeks) showed a reducing trend from week 0 up to the last follow-up. Certain proinflammatory markers, such as CXCL 9 (median concentration of 201.49 pg/mL before and 440.7 pg/mL at 10 weeks), ST2 (median concentration of 1396.27 pg/mL before and 1342.58 pg/mL at 10 weeks), CXCL 11 (median concentration of 89.91 pg/mL before and 491.77 pg/mL at 10 weeks), and IL-Ra (median concentration of 247.74 pg/mL before and 237.63 pg/mL at 10 weeks) tended to be upregulated before infusion and after the four doses of MSC infusions at the 8-week follow-up. Furthermore, we compared nine selected markers between the patients who responded and the patients who did not respond to MSC therapy at the last-follow-up (Figure 4). While there was no statistically significant difference between the responders and non-responders, CXCL 11 (median concentration of 68.77 pg/mL in the responders and 450.49 pg/mL in the non-responders) and IL-Ra (median concentration of 64.25 pg/mL in the responders and 487.68 pg/mL in the non-responders) were significantly lower from the point of the MSC infusion in the responders. In addition, we noted a decline in SPP1 (median concentration of 11,834.15 pg/mL before and 6966.93/mL at 8 weeks), ST2 (median concentration of 1659.08 pg/mL before and 737.89/mL at 8 weeks), and IL-1Ra (median concentration of 64.25 pg/mL before and 0 pg/mL at 8 weeks) immediately after the MSC therapy in the responders. However, these markers showed a subsequent increase after 10 weeks of treatment, underscoring the potential necessity for additional MSC infusions.

### 2.5. Immune Subset Analyses 

We conducted a serial analysis of major immune cell subsets using flow cytometry throughout the follow-up period (Appendix A). Notably, we did not observe any significant differences in the immune cell subsets, including naïve and memory T and B cells. In contrast to our prior findings from in vitro and preclinical investigations, the MSCs did not induce in vivo Treg expansion. However, it could be hypothesized that there was no significant immune suppression despite multiple MSC infusions. Nevertheless, the immune cell subset levels did not necessarily correlate with the clinical response. 

## 3. Discussion

In the present study, we successfully administered repeated infusions of MSCs to ten patients diagnosed with chronic graft-versus-host disease (cGVHD). All the patients received the scheduled four doses of MSCs every two weeks, which were obtained from two bone marrow donors. Notably, clinical efficacy was observed in 6 out of 10 patients, including two cases of complete response resulting in the discontinuation of all immunosuppressive agents. Other responders did not increase immunosuppressives either in dose or therapeutics. This favorable clinical outcome was associated with a reduction in pro-inflammatory cytokines in the serum. However, no statistically significant changes were detected in the absolute numbers of immune cell subsets over the study period.

In recent years, a number of second-line agents for graft-versus-host disease (GVHD) have come to the forefront and received both regulatory and scientific endorsements. These include Janus kinase inhibitors, tyrosine kinase inhibitors, and extracorporeal photopheresis [2]. Indeed, MSC therapy has been considered a therapeutic approach for GVHD over the past two decades, owing to its initial success, especially in the treatment of aGVHD. Despite its popularity and success, MSC therapy failed a phase III sponsor-initiated trial in 2009, but it showed improvement in the liver GVHD group [30]. Through these sub-group analyses, however, Canada and New Zealand approved MSC therapy for pediatric aGVHD patients in 2012, and, subsequently, Japan conditionally approved MSCs for pediatric aGVHD patients in 2015 [31]. Remarkably, MSC therapy has yet to secure approval from the Food and Drug Administration (FDA) in the United States [32,33]. This delay in commercial development despite numerous clinical studies highlights the controversies associated with demonstrating the efficacy of MSCs in the clinic. The application of MSCs in cGVHD has been far less prominent. Nonetheless, studies have shown benefits of MSCs in cGVHD patients despite limited numbers of patients and variability in clinical design.

Multiple factors can contribute to therapeutic potency, including MSC donor variation, cell dose, the timing of administration from cGVHD diagnosis, and the number of doses. In our study, we were able to limit donor variation and cell dose by using MSCs from only two donors. Five patients received MSCs from one donor, and the other five patients received MSCs from the other donor. There were no differences in clinical outcomes between the patients who received MSCs from different donors. Recently, an off-the-shelf donor-pooled MSC product has received approval in India for critical limb ischemia to reduce donor variation while establishing large cell numbers [34]. In the present study, we were successful in administering the same dose and number of doses to all patients. Although prospective parallel studies have not been performed between single and multiple doses, we hypothesized that infusing repeated doses of MSCs could overcome some of the limitations that MSCs have exhibited. In the setting of cGVHD, a range of 1 to 9 doses of MSCs were given at 0.2 to 2.3 × 10^6^ cells/kg [14]. Perez-Simon et al. administered one dose ranging from 0.2 to 1.01 × 10^6^ cells/kg in four out eight patients, two doses in three patients, and four doses in one patient [35], suggesting more of an ‘ad hoc’ administration strategy. One patient achieved complete remission, three showed partial response, and the remaining four patients did not respond. Hermann et al. planned to give eight infusions of MSCs twice weekly for 4 weeks in refractory cGVHD patients [36], but they were only successful in four out of seven patients. In these studies, however, the additional doses did not necessarily correlate with better clinical outcomes, but multivariable factors may be accountable. Conversely, similarly to our study, Boberg et al. were able to successfully infuse at least six up to nine scheduled infusions of MSCs at the same dose [37]. Multiple infusions of MSCs have demonstrated a sustained clinical response, suggesting the importance of ongoing and repetitive MSC therapy. Six out of eleven patients showed an overall partial response at the end of treatment, and three patients showed a mixed response, with the progression of symptoms in some organs. Although different strategies to augment the immunomodulatory capabilities of MSCs by priming or manipulating the MSC product have been extensively explored in preclinical research, they have not yet yielded sufficient clinical evidence. Instead, focusing on refining the clinical approach by adjusting the number of administered doses to maintain in vivo immunomodulatory effects may prove more effective. The practice of repeated MSC infusions remains relatively uncommon and warrants further investigation through future studies.

In addition to multiple doses, the intravenous administration method has been controversial, as MSCs are rapidly cleared in vivo after persistence in the lungs. However, it is now widely perceived that MSCs exhibit therapeutic efficacy through the release of paracrine factors rather than tissue replacement through differentiation. Thus, systemic infusion may be more appropriate for therapy in cGVHD. In a phase III randomized trial [32,33], MSCs failed to show efficacy in aGVHD patients overall. However, they were specifically beneficial in gastrointestinal and pediatric cGVHD patients. In a multicenter trial, 16 patients, including 4 pediatric patients, received multiple infusions of either bone-marrow- or adipose-tissue-derived MSCs. The overall response rate for the pediatric patients was 75%, which was higher than the 58.3% response rate for the adults [38]. Although only a small number of patients were included, these results confirm previous studies in that children achieve better responsiveness than adults. Our study did not include gastrointestinal or pediatric cGVHD patients. Through individual organ assessments, we observed that patients showed clinical improvement in organs that involved the mucosa, such as the skin, eye, and mouth. Thus, specific patient populations may differentially benefit from MSC therapy, and this should be further examined in future studies. 

Moreover, biomarker analyses are becoming increasingly important in clinical studies, as they can help to determine indications for therapy and response to therapy [39,40]. In the Applied Biomarker in Late Effects of Childhood Cancer study, which evaluated the immune profiles of chronic GVHD, increased activated T cells and decreased memory Tregs, along with increased ST2 and soluble CD13, were observed [40]. In preclinical studies, MSCs have clearly shown their ability to induce Tregs both in vitro and in vivo. These observations have been made in several clinical trials; however, in our study, we were unable to see any changes in Treg levels. Previously, we performed a case study of low-dose IL-2 in UPN-4 [41]. In this patient, we saw a significant upregulation of IL-2 throughout the study; however, the expansion in Tregs did not correlate with clinical outcome, as this patient failed to show clinical improvement and was thus enrolled in the current study. Although this patient, as well as other patients, did not show an increase in Tregs or other immune cell subsets, the patients confirmed clinical improvement in self-assessments, as well as in our clinical evaluations. According to these results, Tregs may not be an appropriate biomarker to predict clinical outcome, and other cytokines or immune subsets need to be further evaluated in future studies. In contrast, our study revealed nine cytokines that tend to be upregulated in cGVHD patients. Some of these cytokines, such as ST-2 [42], SPP1 [43], CXCL 9 [43,44], and CXCL 10 [45], are well-known pro-inflammatory cytokines that have been previously described in GVHD. Through the prospective serial monitoring of these selected cytokines, we were able to observe a correlation between clinical improvement and a reduction in pro-inflammatory cytokines. It is also important to note that there may be a difference in the changes in cytokines between responders and non-responders (Figure 4). Although our results do not show statistical significance, markers such as SPP1, CXCL 9, ST2, CXCL 11, and IL-Ra remained elevated at 8 weeks following MSC treatment in the non-responders, suggesting that MSC therapy was insufficient to control the clinical course. However, our study is limited in that only a small number of samples were tested, and the biomarker levels in the peripheral blood may not necessarily reflect the changes in the immune subsets in the tissue or lymph nodes. Therefore, additional studies are needed in the future to guide and help prospectively identify responders versus non-responders to MSC therapy, as this may indicate those in need of additional MSC administrations. 

Newly approved immunomodulatory agents, such as ibrutinib [9] and ruxolitinib [8], are currently being incorporated into second-line management strategies for chronic graft-versus-host disease (cGVHD) [46]. There is a lack of prospective and retrospective studies comparing MSC therapy with other immunosuppressants. Notably, in the REACH3 trial, ruxolitinib demonstrated a 49.7% overall response rate and a 76.4% best overall response at week 24 [8]. In a phase Ib/II multicenter trial, ibrutinib exhibited an overall response rate of 67% at a median follow-up of 13.9 months, with 71% of responders maintaining a sustained response for more than 20 weeks [9]. However, a common observation in both trials was that more than half of the patients who initiated therapy had to discontinue treatment due to adverse events. These significant adverse events encompassed infectious complications, thrombocytopenia, and anemia. In contrast, MSC therapy may offer a more favorable safety profile with fewer adverse events while providing a comparable clinical response, suggesting an advantageous aspect of repeated MSC therapy over kinase inhibitors. 

In conclusion, we show the clinical advantages of intentional multiple infusions of MSCs, with an impressive 80% overall response rate at 8 weeks and a sustained 60% overall response at the 1-year follow-up. There is broad consensus regarding the safety and good tolerance of MSCs in patients, which highlights the need to establish repeated regular administrations of MSCs over a prolonged period of time. To enhance the comparability between studies and to address the limitations associated with MSCs, it is imperative to standardize not only the cell dosage but also the number of infusions in future prospective studies. 

## 4. Patients and Methods

### 4.1. Patients

In this prospective, open-label phase I study, patients aged ≥1 up to ≤80 years with steroid-refractory or progressive cGVHD after allogeneic hematopoietic stem cell transplantation were recruited. The diagnosis of cGVHD was made according to the 2014 National Institutes of Health (NIH)-defined criteria. Steroid-refractory or progressive cGVHD patients were eligible to participate in the study. Steroid-refractory disease was defined as cGVHD without clinical improvement despite ≥4 weeks of standard treatment including corticosteroids and immunosuppressants. Progressive cGVHD was defined as the worsening of symptoms despite ≥2 weeks of standard treatment or during immunosuppressant tapering [47]. 

All patients received corticosteroid therapy prior to MSC infusion. The initiation of new cGVHD medications, concomitant use of other immunosuppressants, and escalation of existing drugs were permitted during the study. In addition to the immunoprophylaxis products listed in Table 1, all patients received low-dose methotrexate at 10 mg/m^2^. Doses of corticosteroids and immunosuppressants could be tapered during the study based on the investigator’s clinical judgement. 

The current study was registered and approved by the Korea Food and Drug Administration on 24 May 2018 (Approval Number: 30860). All patients signed an informed consent form approved by the Institutional Review Board of Seoul St. Mary’s Hospital, the Catholic University of Korea, and the Declaration of Helsinki protocols were followed. Furthermore, the study was registered and approved by the Clinical Research Information Service, Republic of Korea, in the WHO Registry Network (KCT0001894).

### 4.2. Preparation of Bone-Marrow-Derived Mesenchymal Stem Cells

Bone-marrow-derived mesenchymal stem cells were manufactured by the Catholic Institute of Cell Therapy at our in-house academic GMP cell manufacturing facility. Human bone marrow aspirates were obtained from the iliac crest of healthy consenting donors of allogeneic hematopoietic stem cells who were aged 20 to 55 years. The isolation and expansion of MSCs were previously described [48]. Briefly, donors were tested for adequate health status, as well as infectious diseases. Mononuclear cells were then isolated from bone marrow aspirates and were expanded in low-glucose Dulbecco’s modified Eagle’s medium (Gibco, Grand Island, NY, USA) supplemented with 20% fetal bovine serum (FBS). The intermediate product was composed of cells in passage 4, which were washed and frozen in cryopreservation media composed of 10% DMSO, fetal bovine serum, and cell culture medium. Prior to treatment, the intermediate products were thawed and stabilized for an additional passage. The final passage at infusion was passage 5. The release of the final product was based on cell morphology, MSC phenotypical markers, viability, cell number, sterility, endotoxin levels, and the presence of mycoplasma and adventitious viruses. The MSCs were confirmed for their positive expression of CD73 and CD90 and their negative expression of CD31, CD34, and CD45. In addition, the MSCs were capable of differentiation into osteoblasts, adipocytes, and chondrocytes (Appendix A). Any surplus MSCs following release tests and characterization were disposed of.

### 4.3. Treatment

The current study was designed to give each patient a total of 4 doses every two weeks at a cell dose of 1 × 10^6^ cells/kg body weight. All MSCs administered to each patient were derived from one donor. The HLA matching degree between the MSCs and the patient was not considered. For administration, the MSCs were added and diluted into a 0.9% sodium chloride 100 mL mini-bag. Cell administration was completed within thirty minutes. 

### 4.4. Response Assessments

The response criteria were based on the 2014 NIH response criteria publication [49]. Overall response was categorized into three groups for interpretation, namely, complete response (CR), partial response (PR), and no response (NR), which included non-response, mixed response (MR), and progression. CR was defined as the resolution of all manifestations in each organ, and PR was defined as improvement in at least 1 organ without progression in any other organ. Non-response, also referred to as unchanged response, was recorded when the criteria for CR, PR, MR, and progression were not met. MR was defined as CR or PR in at least 1 organ accompanied by progression in another organ. To evaluate progression, each organ must be assessed. Overall, a worsening of 1 point of more in at least one organ without improvement in any other organ was considered progression.

### 4.5. Sample Collection

Blood samples for immune cell profiling and serum analyses were collected from each patient before MSC infusion and at every visit during follow-up. Mononuclear cells were isolated using density gradient centrifugation within the same day of collection and were cryopreserved for later use. Similarly, serum samples were isolated and stored at −20 °C until analysis.

### 4.6. Cell Phenotyping

The cryopreserved peripheral blood mononuclear cells (PBMCs) were thawed in media consisting of RPMI 1640 (Gibco, Grand Island NY, USA) with 10% FBS. The PBMCs were immunostained using various combinations of the following fluorescence-conjugated antibodies (eBioscience): CCR7 (3D12), CD3 (UCHT1), CD4 (SK3), CD8 (SK1), CD11c (3.9), CD14 (61D3), CD16 (CB16), CD19 (HIB19), CD21 (HB5), CD27 (o323), CD38 (HIT2), CD25 (BC96), CD33 (HIM3-4), CD45 (2D1), CD45RA (HI100), CD45RO (UCHL1), CD49b (P1H5), CD56 (TULY56), CD62L (DREG-56), CD66b (G10F5), CD127 (RDR5), CXCR3 (G025H7), HLA-DR (L243), IFN-γ (4S.B3), IgM (G20-127), IL-4 (8D4-88), IL-10 (JES3-9D7), IL-17 (DECEC1), Lag3 (3DS223H), and TNF-α (Mab11). 

For an intracellular cytokine analysis, the PBMCs were stimulated with 25 ng/mL of phorbol myristate acetate (Sigma-Aldrich, St. Louis, MO, USA), 250 ng/mL of ionomycin (Sigma-Aldrich, St. Louis, MO, USA), and 1 μL/mL GolgiSTOP (BD Pharmingen, San Diego, CA, USA) in an incubator with 5% CO_2_ at 37 °C for 4 h. The staining of intracellular cytokines was conducted using an intracellular staining kit (eBioscience, San Diego, CA, USA) according to the manufacturer’s protocol, and they were incubated with anti-IFN-γ, IL-4, IL-10, IL-17, and TNF-α.

For a regulatory T cell (Treg) analysis, surface-stained cells were processed with fixation and permeabilization buffer (eBioscience, San Diego, CA, USA) according to the manufacturer’s protocol and were incubated with anti-Foxp3 (clone PCH101, eBioscience, San Diego, CA, USA).

Data acquisition and analysis were carried out using a fluorescence-activated cell sorting (FACS) LSR Fortessa flow cytometer (BD Biosciences, Bedford, MA, USA) and Flow Jo software version 10.5 (TreeStar, Ashland, OR, USA).

### 4.7. Serum Analysis 

The serum cytokine levels of TNF-α, IL-6, IL-1β, IL-8, sIL-2R, IL-1Ra, IL-4, IL-2, IFN-γ, IL-10, IL-15, IL-12, IL-21, IL-23, IL-7, IL-17A, IL-9, ST2, CXCL11, secreted phosphoprotein 1 (SPP1), peptidase inhibitor 3 (PI3), CXCL9, TGF-β, MMP2, TNF superfamily member 13b (TNFSF13B), CXCL10, MMP3, IL-33, and IL17F were assessed with a Luminex MAGPIX instrument (Luminex, Austin, TX, United States) using custom-made ProcartaPlex Human multiplex immunoassay kits (Affymetrix, Santa Clara, CA, United States) according to the manufacturer’s instructions. Plates were read using a MAGPIX instrument with xPONENT 4.2 software (Luminex, Austin, TX, United States). Cytokine concentrations were calculated using ProcartaPlex Analyst 1.0 software (Affymetrix, Santa Clara, CA, United States). 

The serum cytokine levels of sCD13 and CD163 were determined using an ELISA with Duoset cytokine assay reagents (Biolegend, San Diego, CA, USA) as per the manufacturer’s protocol.

Serum HMGB1 levels were determined using an ELISA with cytokine assay reagents (IBL International GmbH, Hamburg, Germany) as per the manufacturer’s protocol.

### 4.8. Detection of Anti-HLA Antibodies

We examined the presence of HLA-DSA using an LSA assay during each follow-up. The LSA assay for HLA-DSA was performed as per the manufacturer’s instructions using Lifecodes LifeScreen Deluxe kits (Tepnel Lifecodes Corp., Stamford, CT, USA) [50]. In brief, microbeads coated with purified HLA class I/II glycoproteins were incubated with 12.5 μL of patient serum in 96-well plates for 30 min. After three washes with a vacuum manifold, the beads were incubated with 50 μL of a 1:10 dilution of R-phycoerythrin-conjugated goat anti-human immunoglobulin G for 30 min. After washing, the test samples were analyzed using the Quick-Type User’s Manual Research Use Only program, version 2.4, of a LABScan100 flow cytometer (Luminex Corp., Austin, TX, USA); both positive and negative controls were included. The positive criterion was an MFI level of >1000.

### 4.9. Statistical Analysis

Overall survival was defined as the duration between the first infusion and death or the final follow-up. Cell subsets and cytokines were compared using Student’s *t* test, and relative levels were compared using the Mann–Whitney test. For time-dependent changes, a paired *t* test with a repeated ANOVA measure was performed. Differences were considered statistically significant when *p* < 0.05.

## Figures and Tables

**Figure 1 ijms-25-06731-f001:**
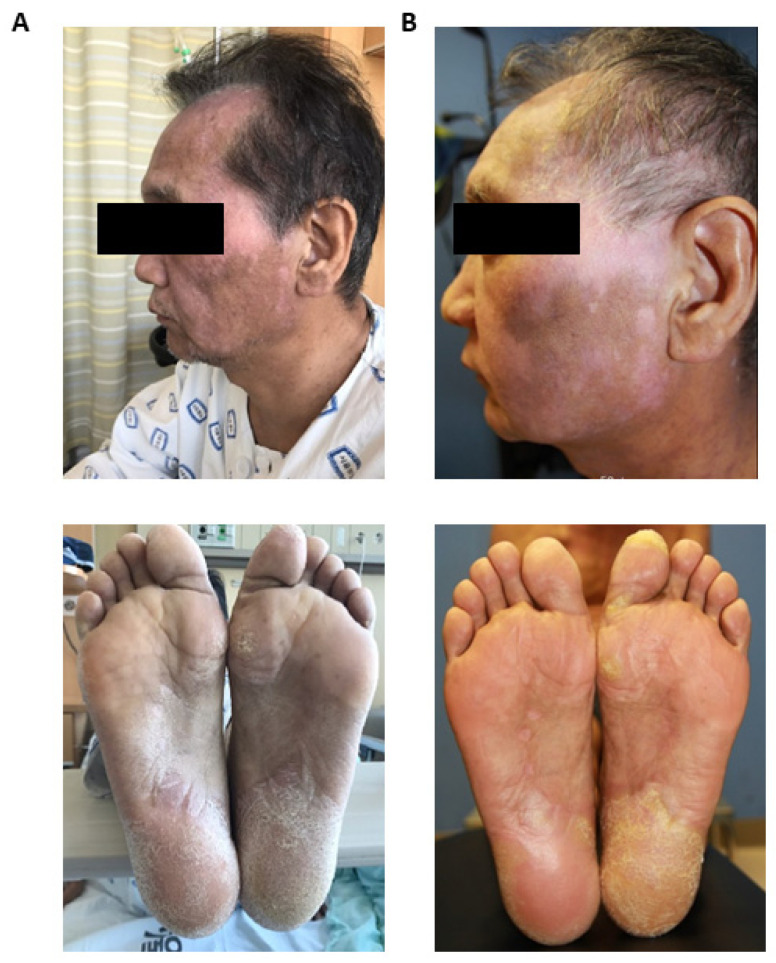
Representative Clinical Improvement of unique patient number (UPN)-1. Face (**upper panel**) and soles (**lower panel**) of the patient prior to (**A**) and following (**B**) four mesenchymal stem cell (MSC) infusions administered every two weeks.

**Figure 2 ijms-25-06731-f002:**
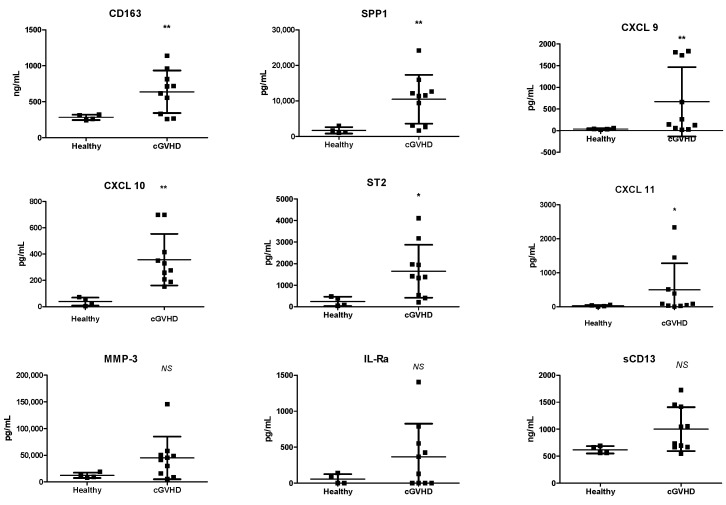
Potential biomarkers upregulated in chronic graft-versus-host disease (cGVHD) patients. Multiplex analysis of plasma samples during the monitoring period is shown. Nine biomarkers that showed significant differences between cGVHD patients (*n* = 10) prior to mesenchymal stem cell (MSC) infusion and healthy controls (*n* = 5) were observed. *p*-values for comparing cGVHD and healthy controls with *t* test: * *p* < 0.05, ** *p* < 0.005. cGVHD, chronic graft-versus-host-disease; IL-Ra, interleukin receptor alpha; CXCL 10, C-X-C motif chemokine ligand 10; CXCL 11, C-X-C motif chemokine ligand 11; CXCL 9, C-X-C motif chemokine ligand 9; MMP-3, matrix metalloproteinase-3; s, soluble; SPP1, secreted phosphoprotein 1; ST2, suppression of tumorigenicity 2.

**Figure 3 ijms-25-06731-f003:**
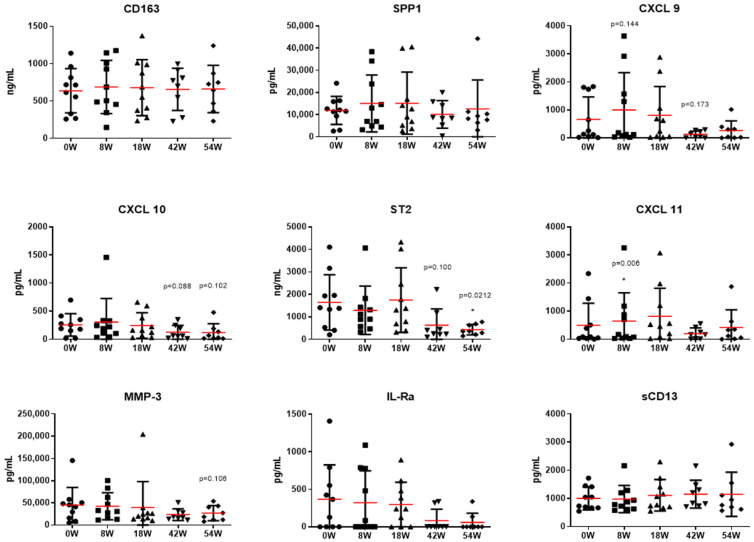
MSC therapy is associated with reduced inflammatory markers. Nine selected biomarkers were observed prior to mesenchymal stem cell (MSC) infusion (0 W) up to 54 weeks. Several inflammatory markers, including CXCL 9, CXCL 10, ST2, and IL-1Ra, showed a reducing trend following MSC infusion. Furthermore, 10 patients were evaluated at weeks 0, 8, and 18; and 8 patients were evaluated at weeks 42 and 54 following the first MSC infusion. Red line indicates mean value, and the error bars indicate the standard deviation. *p*-value for comparing week 0 with *t* test: * *p* < 0.05. IL-Ra, interleukin receptor alpha; CXCL 10, C-X-C motif chemokine ligand 10; CXCL 11, C-X-C motif chemokine ligand 11; CXCL 9, C-X-C motif chemokine ligand 9; MMP-3, matrix metalloproteinase-3; SPP1, secreted phosphoprotein 1; ST2, suppression of tumorigenicity 2; W, week.

**Figure 4 ijms-25-06731-f004:**
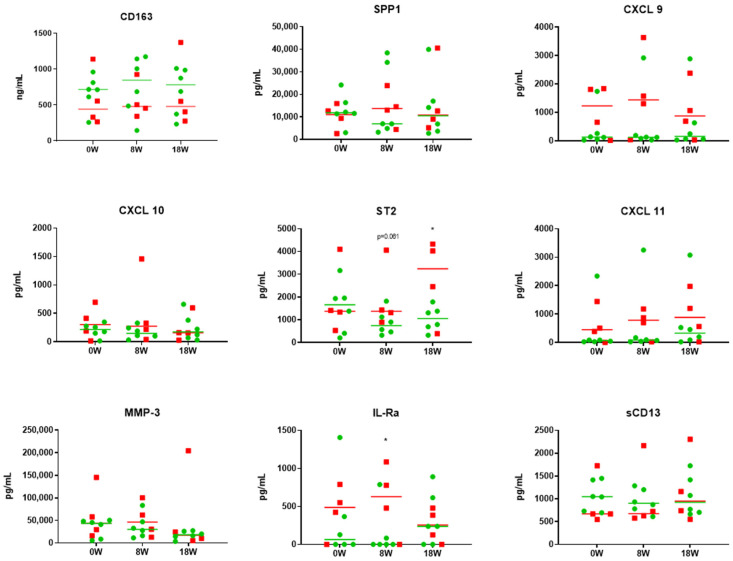
Inflammatory markers are upregulated in non-responders. Nine selected biomarkers were observed prior to mesenchymal stem cell (MSC) infusion (0 W), at 8 and 18 weeks following MSC infusion between responders (*n* = 6) and non-responders (*n* = 4). Green circles and lines indicate patients who responded to MSC therapy. Red squares and lines indicate patients who did not respond to MSC therapy. *p*-value for comparing responders and non-responders with *t* test: * *p* < 0.05. IL-Ra, interleukin receptor alpha; CXCL 10, C-X-C motif chemokine ligand 10; CXCL 11, C-X-C motif chemokine ligand 11; CXCL 9, C-X-C motif chemokine ligand 9; MMP-3, matrix metalloproteinase-3; SPP1, secreted phosphoprotein 1; ST2, suppression of tumorigenicity 2; W, week.

**Table 1 ijms-25-06731-t001:** Patient characteristics.

Patient (UPN#)	Sex/Age	Diagnosis	Conditioning Regimen	GVHD Prophylaxis	HCT Donor *	Months from HCT to cGVHD Diagnosis	Previous Lines of cGVHD Therapy	cGVHD Involvement at pre-MSC Infusion ^+^	cGVHD Score	Overall Severity
1	M/51	Diffuse large B cell lymphoma	Flu6Mel1 TBI800	Cyclosporin	MSD	4.9	3	Oral (3), skin (1), musculoskeletal (2)	6	Severe
2	F/40	Anaplastic large-cell lymphoma	Flu6Mel1 TBI800	Cyclosporin	MSD	6.0	2	Oral (1), eye (2), liver (1), lung (1)	5	Severe
3	M/22	B cell lymphoblastic leukemia	Eto2Cy2 TBI1200	Cyclosporin	MSD	10.5	2	Oral (1), eye (2), liver (1)	4	Moderate
4	M/68	Myelodysplastic syndrome	Flu5Bu2 TBI400	Tacrolimus	MUD	17.1	5	Oral (1), skin (1), eye (1), musculoskeletal (2)	5	Severe
5	F/53	Extranodal NK T cell lymphoma	Flu6Mel1 TBI800	Tacrolimus	MUD	6.0	4	Oral (2), eye (3), musculoskeletal (1)	6	Severe
6	M/52	Extranodal NK T cell lymphoma	Eto2Cy2 TBI1200	Tacrolimus	MUD	6.6	3	Oral (2), eye (2)	4	Moderate
7	M/43	Angioimmunoblastic T cell lymphoma	Flu6Mel1 TBI800	Tacrolimus	MUD	11.1	2	Oral (1), eye (1)	3	Moderate
8	M/31	Myelodysplastic syndrome	Flu5Bu2 TBI800	Tacrolimus	MUD	13.4	4	Oral (1), eye (1) musculoskeletal (2)	4	Severe
9	F/28	Acute myeloid leukemia	Flu5Bu2 TBI400	Tacrolimus	MUD	7.2	5	Skin (1), musculoskeletal (2)	3	Moderate
10	M/17	Myelodysplastic syndrome	Flu4Bu4	Cyclosporin	MSD	7.6	5	Oral (3), skin (3), eye (2), musculoskeletal (1)	9	Severe

Abbreviations: Bu, busulfan; cGVHD, chronic graft-versus-host disease; Cy, cyclophosphamide; Eto, etoposide; F, female; Flu, fludarabine; GVHD, graft-versus-host disease; HCT, hematopoietic cell transplantation; M, male; Mel, melphalan; MSC, mesenchymal stem cell; MSD, matched sibling donor; MUD, matched unrelated donor; NK, natural killer; TBI, total body irradiation; UPN, unique patient number. * HLA antigen compatibility between recipients and MUD donors was 10/10. ^+^ Individual organ scores are indicated in parentheses.

**Table 2 ijms-25-06731-t002:** MSC infusions.

Patient (UPN#)	MSC Lot Number	MSC Day after HCT	MSC HLA Match with Recipient	Anti-HLA Antibodies					
				Class I		Class II		MIC	
				0 W	10 W	0 W	10 W	0 W	10 W
1	BM046SS34	D + 700	0/8	−	−	−	+	−	−
2		D + 607	0/8	−	−	−	−	−	−
3		D + 488	1/8	−	−	−	−	−	−
4		D + 3048	3/8	+	+	+	+	−	−
5		D + 934	1/8	−	−	−	−	−	−
6	BM047SS35	D + 379	0/8	−	+	−	−	+	+
7		D + 605	1/8	−	+	+	+	−	−
8		D + 4449	6/8	+	+	+	+	−	−
9		D + 3334	3/8	−	−	−	−	−	−
10		D + 3393	0/8	−	−	−	−	−	−

Abbreviations: HCT, hematopoietic cell transplantation; HLA, human leukocyte antigen; MSC, mesenchymal stem cell; UPN, unique patient number.

**Table 3 ijms-25-06731-t003:** Clinical outcomes.

Patient (UPN#)	Initial Response at 8 W	Response at 18 W	Response at Last F/U	Outcome
1	PR	PR	PR	Alive
2	PR	PR	PR	Alive
3	NR	PR	CR	Alive
4	PR	PR	PR	Alive
5	PR	NR	NR *	Died—Relapse
6	PR	MR	MR *	Died—Relapse
7	PR	PR	MR	Alive
8	MR	MR	MR	Alive
9	PR	PR	PR	Alive
10	PR	PR	CR	Alive

* At 18 weeks. Abbreviations: CR, complete response; F/U, follow-up; MR, mixed response; NR, no response; PR, partial response; W, weeks; UPN, unique patient number.

**Table 4 ijms-25-06731-t004:** Individual organ response.

Patient (UPN#)	Organ Scoring and Response																											
	Skin				Mouth				Eyes				GI Tract				Liver				Lungs				Joints/Fascia			
Weeks after MSC Treatment	0	8	18	L	0	8	18	L	0	8	18	L	0	8	18	L	0	8	18	L	0	8	18	L	0	8	18	L
1	1	1	1	0	3	2	3	1	0	0	0	0	0	0	0	0	0	0	0	0	0	0	0	0	2	1	0	0
2	0	0	0	0	1	0	0	0	2	1	1	1	0	0	0	0	1	1	1	1	1	1	1	1	0	0	0	0
3	0	0	0	0	1	1	1	0	2	2	0	0	0	0	0	0	1	0	0	0	0	0	0	0	0	0	0	0
4	1	0	1	1	1	0	0	0	1	1	0	0	0	0	0	0	0	0	0	0	0	0	0	0	2	1	1	1
5	0	0	0	0	2	2	2	2	3	2	3	3	0	0	0	0	0	0	0	0	0	0	0	0	1	1	1	1
6	0	0	1	1	2	0	1	1	2	1	0	0	0	0	0	0	0	0	0	0	0	0	0	0	0	0	0	0
7	0	0	0	1	2	1	1	1	1	1	0	0	0	0	0	0	0	0	0	0	0	0	0	0	0	0	0	1
8	0	1	1	1	1	0	0	0	1	1	1	1	0	0	0	0	0	0	0	0	0	0	0	0	2	1	2	2
9	1	1	1	1	0	0	0	0	0	0	0	0	0	0	0	0	0	0	0	0	0	0	0	0	2	1	1	1
10	3	2	1	1	3	1	1	1	2	1	1	1	0	0	0	0	0	0	0	0	0	0	0	0	1	0	0	0
ORR at 8 W	2/6 (33%)				7/9 (78%)				4/8 (50%)				0/0				1/2 (50%)				0/1 (0%)				5/6 (83%)			
ORR at 18 W	1/6 (16%) 2 PD				6/9 (66%)				6/8 (75%)				0/0				1/2 (50%)				0/1 (0%)				4/6 (67%)			
ORR at last F/U	2/6 (33%) 3 PD				8/9 (89%)				6/8 (75%)				0/0				1/2 (50%)				0/1 (0%)				4/6 (67%) 1 PD			

Abbreviations: F/U, follow-up; GI, gastrointestinal; L, last visit; ORR, overall response rate; PD, progressive disease; UPN, unique patient number.

**Table 5 ijms-25-06731-t005:** Immunosuppressants used during the study.

Patient (UPN#)	Immunosuppressive Agents before MSC Infusion		Immunosuppressive Agents at the Last Follow-Up	
	PSL, mg	Other Agent	PSL, mg	Other Agent
1	10	Cyclosporine	10	Cyclosporine
2	10	Cyclosporine, MMF	10	Cyclosporine, MMF
3	15	Cyclosporine	0	None
4	7.5	MMF	0	Repaglanide
5	10	Tacrolimus	10	Tacrolimus
6	7.5	MMF, Tacrolimus	7.5	None
7	10	MMF, Tacrolimus	0	Repaglanide, Tacrolimus
8	10	-	0	None
9	5	Tacrolimus	0	None
10	-	AZA, Hydroxychloroquine Sulfate, Repaglanide	0	None

Abbreviations: AZA, azathioprine; MMF, mycophenolate mofetil; PSL, prednisolone; UPN, unique patient number.

## Data Availability

No new data were created or analyzed in this study. Data sharing is not applicable to this article.

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
