# Peer review of "Repeated Infusions of Bone-Marrow-Derived Mesenchymal Stem Cells over 8 Weeks for Steroid-Refractory Chronic Graft-versus-Host Disease: A Prospective, Phase I/II Clinical Study"

_ijms, 2024, doi:10.3390/ijms25126731_

Round 1

Reviewer 1 Report

Comments and Suggestions for Authors

Comments on the Quality of English Language

Moderate editing of English language with extensive formatting needed.

Author Response

The responses are also attached as a separate file.

Response to the comments by Reviewer #1

- I recommend that the title specifies the duration of treatment rather than “repeated infusion”.

[Response] We revised the title as follows to include the duration of treatment:

“Repeated Infusions of Bone-Marrow Derived Mesenchymal Stem Cells Over 8 Weeks for Steroid-Refractory Chronic Graft-versus-Host Disease: a Prospective, Phase I/II Clinical Study”

- The authors should shed more light on previous clinical trials using MSCs in different diseases,with the response rate and incidence of side effects; particularly in acute and Chronic Graft-versus-Host Disease.

[Response] In this study, we focus particularly on the use of MSCs in acute and chronic graft-versus-host disease. Below we summarize the response rate of MSCs in steroid refractory aGVHD and cGVHD, summarized from Front Immunol. 2021; 12: 761616. MSCs are generally safe without side effects and well tolerated. However, the response rates vary depending on the dose regimens, study designs, disease status, etc.

Ref

n

MSC dose (cells/kg)

# of infusions

CR

OR

OS

Treatment of steroid refractory aGVHD

Von Bonin et al.

13

0.6-1.1x106

1-5

8% (D28)

54% (D28)

Not specified

Lucchini et al.

8

0.7-1.6x106

1-2

24% (D28)

71% (D28)

63% (1y)

Muroi et al.

14

2x106

8-12

75% (D28)

93% (D28)

57% (2y)

Introna et al.

40

1x106

>2

28% (D28)

68% (D28)

50% (1y)

38% (2y)

Zhao et al.

47

1x106

2-8

36% (D28)

75% (D28)

45% (3y)

Muroi et al.

25

2x106

8-12

24% (D28)

60% (D28)

48% (1y)

Salmenniemi et al.

26

2x106

1-6

27% (D28)

62% (D28)

Not specified

Bader et al.

69

1-2x106

1-4

32% (D28)

83% (D28)

71% (6m)

Kebriaei et al.

260

2x106

8-12

37% (D28)

58% (D28)

34% (D180)

Kurtzberg et al.

241

2x106

8-12

14% (D28)

65% (D28)

67% (D100)

Kurtzberg et al.

54

2x106

8-12

30% (D28)

70% (D28)

69% (D180)

Prasad et al.

12

2 or 8x106

8-12

17% (D32)

58% (D60)

67% (D32)

75% (D60)

58% (D100)

40% (2y)

Sanchez-Guijo et al.

24

0.7-1.3x106

2-4

46% (D60)

71% (D60)

44% (1y)

Ringden et al.

8

0.7-9x106

1-2

Not specified

Not specified

38% (2y)

Le Blanc et al.

55

0.4-9x106

1-5

Not specified

Not specified

35% (2y)

Arima et al.

3

0.5x106

1

Not specified

Not specified

0% (2y)

Perez-Simon et al

10

0.6-2.9x106

1-4

Not specified

Not specified

Not specified

Hermann et al.

12

1.7-2.3x106

2-19

Not specified

Not specified

50% (3 y)

Ball et al.

37

0.9-3x106

1-19

Not specified

Not specified

Not specified

Resnick et al.

50

0.3-2.3x106

1-4

Not specified

Not specified

Not specified

Von Dalowski et al.

58

0.5-2.1x106

1-6

Not specified

Not specified

19% (1y)

17% (2y)

Dotoli et al.

46

1-29.8x106

1-7

Not specified

Not specified

20% (1y)

17% (2y)

Treatment of cGVHD

Ringden et al.

1

0.6x106

1

No response

No response

0% (1y)

Lucchini et al.

5

0.7-1.4x106

1-4

40% (D28)

80% (D28)

Time frame not specified

Perez Simon et al.

10

0.2-1.2x106

1-4

Time frame not specified

Time frame not specified

Time frame not specified

Hermann et al.

7

1.7-2.3x106

2-11

Time frame not specified

Time frame not specified

29% (1y)

Jurado et al.

14

1 or 3x106

1

57% for low dose (1y)

80% for high dose (1y)

67% for low dose (1y)

80% for high dose (1y)

67% (1y)

Salmenniemi et al.

4

2x106

1-6

No response

No response

25% (3m)

Boberg et al.

11

2x106

6-9

Not reported

Time frame not specified

Time frame not specified

Table is adapted from Front Immunol. 2021; 12: 761616.

In the manuscript, we added more information of previous clinical studies including response rates and side effects when applicable:

“Perez-Simon et al. administered one dose ranging from 0.2 to 1.01 x 106 cells/kg in 4 out 8 patients, two doses in 3 patients, and 4 doses in one patient [24] suggesting more of an ‘ad hoc’ administration strategy. One patient achieved complete remission, three showed partial response, while the remaining four patients did not respond. Hermann et al. planned to give 8 infusions of MSCs twice weekly for 4 weeks in refractory cGVHD patients [25], but was only successful in 4 out of 7 patients. In these studies, however, the additional doses did not necessarily correlate with better clinical outcomes, but multivariable factors may be accountable. On the other hand, similar to our study, Boberg et al. were able to successfully infuse at least six up to nine scheduled infusions of MSCs at the same dose [26]. Multiple infusions of MSC treatment have demonstrated a sustained clinical response, suggesting the importance of ongoing and repetitive MSC therapy. Six out of eleven patients showed an overall partial response at the end of treatment and three patients showed a mixed response with progression of symptoms in some organs.”

“In a multicenter trial, 16 patients including 4 pediatric patients received multiple infusions of either bone marrow or adipose tissue derived MSCs. The overall response rate for the pediatric patients was 75% which was higher than the 58.3% response rate for the adult [27]. Although only a small number of patients was included, these results confirm previous studies that children achieve better responsiveness compared to adults.”

- On what basis was sample size selected (10 patients)?

[Response] Due to funding and feasibility constraints, the sample size was limited to 10 patients. Therefore, this is an exploratory Phase I/II trial evaluating the safety and preliminary efficacy of MSCs in cGVHD. 

- Were there any exclusion criteria?

[Response] The exclusion criteria was as follows:

  • Pregnant or breastfeeding women, women who may become pregnant, or those not taking appropriate contraceptive measures.
  • Patients with active infections or fever (≥38℃) of unknown etiology or ongoing bacterial or fungal infections.
  • Pediatric or elderly patients under 1 year of age or over 80 years of age.
  • Patients with an Eastern Cooperative Oncology Group (ECOG) performance status of 3 or 4.
  • Patients with previously documented HIV infection, uncontrolled hypertension (diastolic blood pressure > 115 mmHg), unstable angina, congestive heart failure (NYHA class II or higher), poorly controlled severe diabetes, coronary angioplasty within the last 6 months, acute myocardial infarction within the past 6 months, or uncontrolled atrial or ventricular arrhythmias including other serious non-malignant conditions.
  • Patients with psychiatric disorders, drug addiction, or other conditions that could impact the study results.
  • Patients currently participating in another clinical trial (unless determined otherwise by the investigator's clinical judgment).
  • Patients deemed unsuitable for the study by the clinical trial investigator (or responsible person).
  • Patients with other serious medical conditions that could reduce compliance with the clinical trial.

- What is the rationale for selecting the dose regimen (cell dosage and number of infusions)?

[Response] The cell dose for bone marrow derived MSCs in cGVHD has ranged from 0.6 x 106 cells/kg up to 3 x 106 cells/kg in previous clinical trials. The dose 1 x106 cells/kg was chosen because it is within the range of previously reported clinical studies. Furthermore, the number of doses has ranged from single infusion to 11 infusions in previous studies. However, the study that gave up to 11 infusions was performed only in one patient out of 7 patients. Our main goal was to give the same dose and same cell dose to all patients, therefore the cell dose and number of infusions had to be feasible for preparing the cells. Therefore, we chose 1x106 cells/kg given every 2 weeks for a duration of 8 weeks.  

Ref

n

MSC dose (cells/kg)

# of infusions

CR

OR

OS

Ringden et al.

1

0.6x106

1

No response

No response

0% (1y)

Lucchini et al.

5

0.7-1.4x106

1-4

40% (D28)

80% (D28)

Time frame not specified

Perez Simon et al.

10

0.2-1.2x106

1-4

Time frame not specified

Time frame not specified

Time frame not specified

Hermann et al.

7

1.7-2.3x106

2-11

Time frame not specified

Time frame not specified

29% (1y)

Jurado et al.

14

1 or 3x106

1

57% for low dose (1y)

80% for high dose (1y)

67% for low dose (1y)

80% for high dose (1y)

67% (1y)

Salmenniemi et al.

4

2x106

1-6

No response

No response

25% (3m)

Boberg et al.

11

2x106

6-9

Not reported

Time frame not specified

Time frame not specified

Table is adapted from Front Immunol. 2021; 12: 761616.

- Please discuss this in light of published clinical trials. What is the rationale for serum biochemical analysis and to what extent are these biomarkers reflective of patients’ response and reflective of the incidence of side effects?

[Response] The cGVHD NIH Consensus Project (Blood Adv. 2023 Sep 12; 7(17): 4886–4902) has emphasized the need for validated biomarkers in cGVHD to improve the precision of diagnosis, treatment monitoring and prognosis. While some progress has been made on biomarkers that may identify patients who are at high risk of developing cGHVD following allogeneic hematopoietic stem cell transplantation, further studies are needed to determine biomarkers that may be associated with response and prognosis.

In the ABLE study (Blood (2020) 135 (15): 1287–1298.), that evaluated the immune profiles in cGVHD patients, decreased transitional B cells along with increased NK cell subsets have been observed. cGVHD had additional abnormalities with increased activated T cells, naïve helper T cells, and loss of CD56 bright NK cells, as well as increased ST2 and soluble CD13 levels. Similarly, in our study ST2 and sCD13 were upregulated markers in chronic graft-versus host disease and showed decrease following MSC treatment (Figure 2). In addition, there was a statistically significant difference in the levels of ST2 in the responders vs. non-responders at 18 weeks of follow-up indicating an association of response (Figure 4).

Furthermore, biochemical markers such as liver enzymes and renal function test may be directly reflective of organ-specific toxicity and side effects from the treatments. However, because liver enzymes tend me upregulated in cGVHD patients, it may also be reflective of response to treatment in liver cGVHD.

In the discussion section on biomarkers, we have added:

“Moreover, biomarker analyses are becoming increasingly important in clinical studies as they can help decide indication for therapy and also response to therapy [27, 28]. In the Applied Biomarker in Late Effects of Childhood Cancer study which evaluated the immune profiles of chronic GVHD, increased activated T cells, decreased memory Tregs along with increased ST2 and soluble CD13 was observed [28].”

- Results Section 3.2 MSC induces long-term clinical response in majority of cGVHD patients, the authors stated that “One patient showed a mixed response rate in which an improvement and deterioration was both observed” please elaborate more.

[Response] We modified the sentence as follows to include more details:

“Patient 8 showed a mixed response at every response assessment in which the patient showed improvement in the oral mucosa and joints, but deterioration in the skin, and also stable disease in the eyes.”

- At the last follow up, which is approximately one year from the first infusion, the overall response rate was 60%, with a complete response rate of 20% and a PR rate of 40%. Is this response rate considered favorable in this group of patients? This should be highlighted in comparison with all published data and clinical trials.

[Response] Below we have compared the therapeutic options available for steroid-refractory cGVHD and their response rates. The results of our study seem to be favorable or within the range of other secondary therapeutic options.

Ref

Type of treatment

Response Rate

OS

Miklos et al.

Walker et al.

Ibrutinib (BTK inhibitor)

CR 21%, PR 45% with median follow-up of 13.9 months

71% at 2 years

Oarbeascoa et al.

Couriel et al.

Extracorporeal photopheresis

CR 23%, PR 44%

53-78% at 1 year

Flowers et al.

Mycophenolate mofetil

OR 24-64%

67-96% at 1 year

Zeiser R et al.

Rituximab

OR 65%

76% at 2 years

Zeiser R et al.

Ruxolitinib

CR 7%, PR 78%

97% at 6 months

Jurado M et al.

mTOR inhibitor

CR 38%, PR 43%

-

Jacobsohn DA et al.

Methotrexate

OR 55%

70% at 2 years

Cutler C et al.

Belumosudil (ROCK2 inhibitor)

CR 3%, PR 71%

FFS 77% at 6 months

Chao NJ et al.

Cyclophosphamide

60% of 15 patients showed improvement after 8-12 monthly cycles

-

Weng et al.

MSCs

CR 21%, PR 53%

78% at 2 years

Current Study

MSCs

CR 20%, PR 60% at 1 year

80% at 1 year

Table is adapted from Bone Marrow Transplant. 2021; 56(9): 2079–2087.

In the discussion section, we commented on the response rate of our data compared to other therapeutic options:

“Newly approved immunomodulatory agents, such as ibrutinib[34] and ruxolitinib[35], are currently being incorporated into second-line management strategies for chronic graft-versus-host disease (cGVHD)[36]. There is a lack of prospective or retrospective studies comparing MSC therapy with other immunosuppressants. Notably, in the REACH3 trial, ruxolitinib demonstrated a 49.7% overall response rate and a 76.4% best overall response at week 24 [35]. In a phase Ib/II multicenter trial, ibrutinib exhibited an overall response rate of 67% at a median follow-up of 13.9 months, with 71% of responders maintaining a sustained response for over 20 weeks [34]. However, a common observation in both trials is that more than half of the patients who initiated therapy had to discontinue treatment due to adverse events. These significant adverse events encompass infectious complications, thrombocytopenia, and anemia. In contrast, MSC therapy may offer a more favorable safety profile with fewer adverse events while providing a comparable clinical response suggesting an advantageous aspect of repeated MSC therapy over kinase inhibitors.

- More references should be included addressing the clinical applications of MSCs in different diseases.

[Response] Thank you for your comments. We have included references on the clinical applications of MSCs in different diseases in the introduction section.

“Clinical applications of MSCs include but are not limited to acute respiratory distress syndrome[16], Crohn’s disease[17], Type 1 diabetes[18], osteoarthritis[19], multiple sclerosis[20], and liver diseases[21].”

- English, grammer and formatting need to be thoroughly revised, including titles of tables.

[Response] Thank you for your comments. We have submitted our paper to MDPI’s English editing service for English editing. 

Reviewer 2 Report

Comments and Suggestions for Authors

Chronic graft-versus-host disease (cGVHD) is a longterm complication after alloHSCT. Steroids are the first line treatment for the disease, however, for same patients how don’t respond to steroid, other treatments are needed.  In this this study, the authors study the effect of Mesenchymal stem cells (MSCs) as potential treatment and evaluate the safety and efficacy of repeated infusions of MSCs. The manuscript and results are important and well presented, however, the manuscript needs few changes to improve its quality.

Major comments

-        In the introduction, as second line treatment, the author need to include more treatment such as, cyclophosphamide, IL-17 (belumosudil) , mTOR-Inh, methotrexate, PDGFα (imatinib, niolotinib)….etc. Also more previous studies are needed.

-        It will be of great interest if you could use matched MSC donors to see if this will improve the outcome. Did you check the compatibility between recipients and MSC donors?

Minor comments

-        In line 26, the typo need correction (1x106 cells/kg)

-        Osteopontin is the previous name, the recent name is secreted phosphoprotein 1 (SPP1), the same for Elafin, the new name is peptidase inhibitor 3 (PI3), Baff: TNF superfamily member 13b (TNFSF13B), MIG: CXCL9, I-TAC: CXCL11…check all other genes (visit https://www.genenames.org/)

-        When mentioning the company, from where you purchased reagents, Kits or devices, the information must be uniform and should include (company, city and country) like in line 178-179.

-        Typos correction needed

Typo in line 92

Typo in line 131 (1x106 cells…)

Typo in line 215 (1x106 cells…)

Typo in line 419 and 420 (1x106 cells…)

Comments on the Quality of English Language

Minor editing of English, need to correct typos

Author Response

Responses are also attached as a separate file.

Major comments:

- In the introduction, as second line treatment, the author need to include more treatment such as, cyclophosphamide, IL-17 (belumosudil) , mTOR-Inh, methotrexate, PDGFα (imatinib, niolotinib)….etc. Also, more previous studies are needed.

[Response] Thank you for your comments. We included other second-line treatments including  belumosudil, mTOR inhibitor, imatinib, nilotinib.

“For the remaining half, there has been multiple drugs and approaches that have been approved as second-line treatment [2, 3] including mTOR inhibitor[4], nilotinib[5], imatinib[6], extracorporeal photopheresis (ECP)[7], ruxolitinib [8], ibrutinib[9, 10] and belumosudil[11].”

- It will be of great interest if you could use matched MSC donors to see if this will improve the outcome. Did you check the compatibility between recipients and MSC donors?

[Response] Thank you for your comments. The compatibility between recipients and MSC donors is mentioned in Table 2. The MSC donor HLA matching degree with the recipient ranged from 0/8 up to 6/8; however, there was no significant difference in the clinical outcome depending on the matching degree. It would be of great interest in future studies to see whether the HLA compatibility between the MSC donor and recipient could improve the clinical outcome.

Minor comments:

- In line 26, the typo need correction (1x106 cells/kg)

[Response] The typo was corrected to 1x106 cells/kg.

- Osteopontin is the previous name, the recent name is secreted phosphoprotein 1 (SPP1), the same for Elafin, the new name is peptidase inhibitor 3 (PI3), Baff: TNF superfamily member 13b (TNFSF13B), MIG: CXCL9, I-TAC: CXCL11…check all other genes (visit https://www.genenames.org/)

[Response] Osteopontin was corrected to secreted phosphoprotein 1 (SPP1), elafin was corrected to peptidase inhibitor 3 (PI3), BAFF was corrected to TNF superfamily member 13b (TNFSF13B) in the manuscript.

In addition, MIG was corrected to CXCL9, I-TAC to CXCL11, and IP-10 to CXCL10.

- When mentioning the company, from where you purchased reagents, Kits or devices, the information must be uniform and should include (company, city and country) like in line 178-179.

[Response] The information of the company where we purchased the reagents has been modified so that the information is uniform.

- Typos correction needed:

- Typo in line 92

[Response] Typo was corrected to m2.

- Typo in line 131 (1x106 cells…)

- Typo in line 215 (1x106 cells…)

- Typo in line 419 and 420 (1x106 cells…)

[Response] Typos in line 131, 215, 419 and 420 were corrected to 1x106 cells
